# Saikosaponin B2, Punicalin, and Punicalagin in Vitro Block Cellular Entry of Feline Herpesvirus-1

**DOI:** 10.3390/v16020231

**Published:** 2024-02-01

**Authors:** Bin Liu, Xiao-Qian Jiao, Xu-Feng Dong, Pei Guo, Shu-Bai Wang, Zhi-Hua Qin

**Affiliations:** College of Veterinary Medicine, Qingdao Agricultural University, Qingdao 266109, China; liubin@stu.qau.edu.cn (B.L.); xqj426@stu.qau.edu.cn (X.-Q.J.); dongxufeng@qau.edu.cn (X.-F.D.); guopei61@163.com (P.G.); 199101020@qau.edu.cn (S.-B.W.)

**Keywords:** feline herpesvirus-1, envelope glycoprotein B, Bio-layer interferometry, natural compounds, virus entry inhibitor

## Abstract

In the realm of clinical practice, nucleoside analogs are the prevailing antiviral drugs employed to combat feline herpesvirus-1 (FHV-1) infections. However, these drugs, initially formulated for herpes simplex virus (HSV) infections, operate through a singular mechanism and are susceptible to the emergence of drug resistance. These challenges underscore the imperative to innovate and develop alternative antiviral medications featuring unique mechanisms of action, such as viral entry inhibitors. This research endeavors to address this pressing need. Utilizing Bio-layer interferometry (BLI), we meticulously screened drugs to identify natural compounds exhibiting high binding affinity for the herpesvirus functional protein envelope glycoprotein B (gB). The selected drugs underwent a rigorous assessment to gauge their antiviral activity against feline herpesvirus-1 (FHV-1) and to elucidate their mode of action. Our findings unequivocally demonstrated that Saikosaponin B2, Punicalin, and Punicalagin displayed robust antiviral efficacy against FHV-1 at concentrations devoid of cytotoxicity. Specifically, these compounds, Saikosaponin B2, Punicalin, and Punicalagin, are effective in exerting their antiviral effects in the early stages of viral infection without compromising the integrity of the viral particle. Considering the potency and efficacy exhibited by Saikosaponin B2, Punicalin, and Punicalagin in impeding the early entry of FHV-1, it is foreseeable that their chemical structures will be further explored and developed as promising antiviral agents against FHV-1 infection.

## 1. Introduction

Feline herpesvirus-1 (FHV-1) is a widespread pathogen in felines that clinically induces severe upper respiratory and ocular signs, including congestive conjunctivitis, respiratory distress, and viral pneumonia [1,2]. There are currently no therapeutic drugs specifically targeting FHV-1 infection, and FHV-1 can be latently infected for life, which makes it extremely difficult to treat and control FHV-1 infection [3].

FHV-1 is a double-stranded DNA virus of the *Herpesviridae* family with a glycoprotein lipid envelope. Several surface glycoproteins have been identified on FHV-1, namely envelope glycoprotein B (gB), gC, gD, gE, gG, gH, gI, gJ, gK, gL, gM, and gN [4,5]. gB is a glycoprotein essential for herpesvirus infection of cells, and it plays a role in viral adsorption and mediation of membrane fusion, which is very important for viral infection of host cells [6]. Recent studies have reported that gB is the most conserved glycoprotein in herpesviruses, and during viral infection of host cells, gB causes viral adsorption to the cell surface by binding to the cellular marker receptor acetyl heparin sulfate; the protein conformation of gB is altered to produce an anchoring-like effect that further drives fusion of the viral envelope with the host envelope [7,8]. In addition, it has been shown that gB also plays an important role in the release of mature herpesvirus particles [9]. Therefore, novel small molecule inhibitors recognizing gB are also important avenues for FHV-1 infection treatment and drug design.

In recent years, therapies against FHV-1 infections have been based on the use of nucleoside analogs, which target the viral DNA polymerase to induce viral gene deletion or inhibit DNA synthesis. However, these drugs were not considered for the treatment of FHV-1 infection during the initial development process, leading to several clinical problems. For example, acyclovir has significant oral toxicity in felines. Similarly, penciclovir, cidofovir, and famciclovir show potent anti-FHV-1 activity in vitro but lack efficacy in alleviating clinical symptoms; moreover, long-term use of these antiviral drugs with a single mechanism of action predisposes the virus to developing drug resistance [10,11,12,13,14]. Therefore, more effective drugs and drugs with different mechanisms are needed to combat FHV-1 infection.

“Small molecule drugs” generally refer to drugs with relatively low molecular weight and relatively simple chemical structures. In this study, given the highly diverse structural and physicochemical characteristics of small molecule drugs, in addition to their multiple potential targets of action [15], we screened small molecule drugs against gB using BLI and tested the antiviral activity of the screened drugs against FHV-1 infection. It has been determined that Saikosaponin B2, Punicalin, and Punicalagin are potent inhibitors against the early stages of FHV-1 infection, especially against the viral attachment and fusion steps. In addition, this study further validates the feasibility of screening for viral entry inhibitors using herpesvirus surface functional glycoprotein gB.

## 2. Materials and Methods

### 2.1. Compounds

Saikosaponin B2, Punicalin, and Punicalagin, along with other compounds, were purchased from MedChemExpress (Shanghai, China). These compounds, boasting a purity exceeding 95%, were accompanied by either NMR or UPLC-UV proof, ensuring their quality and authenticity. Before experimentation, the drugs were diligently stored at −80 °C to maintain their stability and integrity.

### 2.2. Chemicals and Reagents

Dimethyl sulfoxide (DMSO), an essential solvent for these compounds, was sourced from Solarbio (Beijing, China). DIOC, R18, and Cell Counting Kit-8 (CCK-8) were all purchased from TargetMol Chemicals Inc. (Shanghai, China). DMEM culture medium was obtained from GIBCO (Shanghai, China). Dulbeccos Phosphate-Buffered (D-PBS) was purchased from Sangon Biotech (Shanghai, China) Co., Ltd. Amicon Ultra-15 centrifugal filter was purchased from Sigma-Aldrich LLC (St Louis, MO, USA).

### 2.3. Cells and Virus

CRFK cells (ATCC:CCL-94) were cultivated in Dulbecco’s Modified Eagle Medium (DMEM, Gibco) supplemented with 10% heat-inactivated fetal calf serum (FCS, Gibco, Shanghai, China), 100 U/mL of penicillin, and 100 μg/mL of streptomycin (Solarbio, Beijing, China). These cells were maintained in a controlled environment at 37 °C with 5% humidified CO_2_.

FHV-1 was isolated from infected cats, stored at −80 °C, and named FHV/BJ-1. Calculated using the Reed–Muench formula, the FHV-1 TCID_50_ is 10^−5.8^/0.1 mL. In the early stages of isolation, it was co-infected with Feline Panleukopenia Virus (FPV), and after bacteriophage purification, a pure strain of FHV-1 was obtained. The gB, gD, gC, gE, gG, gH, gI, and gL envelope protein genes of the obtained FHV-1 strain were amplified and sequenced. Amino acid variation analysis and phylogenetic evolution analysis were conducted on the sequencing results. This study revealed that, compared to the first isolated strain C-27 (FJ478159.2), the BJ-1 strain had only one common nonsynonymous mutation in the gI gene, specifically, a T mutation at the 165th amino acid of the gI protein. Simultaneously, regression experiments conducted in animal bodies indicated that the BJ-1 strain is highly virulent, causing clinical symptoms in cats inoculated with this strain. For all virus infection experiments, a basal medium containing 2% fetal bovine serum (FBS) with 1% antibiotics was employed.

### 2.4. Expression and Purification of gB

Bacmid vectors were meticulously crafted through homologous recombination via the transformation of the pFastBac vector plasmid, incorporating the FHV-1 gB sequence into the DH10Bac strain. The resulting Bacmid vector was then isolated and transfected into SF21 insect cells, which were cultivated under optimal conditions for subsequent expression of the FHV-1 gB protein.

Utilizing the AKTA protein purification system, we efficiently gathered the desired proteins. Subsequently, the accuracy of the proteins was confirmed through Western blot analysis. The protein’s molecular weight was confirmed at 59 kDa, and its purity reached 94% with a protein concentration of 3 mg/mL.

### 2.5. Biolayer Interferometry (BLI) Assay

The immobilization of FHV-1 gB was carried out on Ni-NTA probes obtained from Fortebio (Fremont, CA, USA). Small molecule drugs were meticulously diluted in phosphate-buffered saline (PBS) to concentrations ranging from 6.25 μM to 100 μM. Initially, a baseline was established using PBS containing 1% DMSO. Subsequently, biosensor tips coated with the protein were immersed into wells containing different concentrations of the small molecule drug dilutions for 60 s to facilitate binding, followed by a separation period of 60 s. The dissociation constant (KD) values were calculated utilizing a 1:1 binding model through Data Analysis Software 9.0, provided by Fortebio.

### 2.6. Cytotoxicity Assay

CRFK cells were seeded at a density of 1 × 10^5^ cells/mL in a 96-well plate, with 100 μL per well. After the cells reached a confluent monolayer, a cell drug toxicity test was conducted. The cells were cultured in a cell maintenance solution (2% FBS) containing different concentrations of the test drug in a cell culture incubator for 48 h to allow sufficient contact between the drug and the cells. After an additional 48-h incubation, the supernatant was removed, and 110 μL of a CCK-8 solution prepared in serum-free medium was added to each well. Subsequently, the plates were incubated for an extra 2 h at 37 °C with 5% CO_2_. Absorbance measurements were performed at a wavelength of 450 nm. The acquired data underwent analysis utilizing GraphPad Prism 8.0 software, enabling the calculation of the 50% cytotoxic concentration (CC_50_). This parameter signifies the drug concentration at which 50% of the cells display cytotoxic effects. Cell viability was calculated using Equation (1):(1)Cell viability%=ODexperimental−ODblankODcontrol−ODblank×100

### 2.7. Cytopathic Effect (CPE) Inhibition Assay

A monolayer of CRFK cells was cultured in a 96-well plate and subsequently inoculated with a cell maintenance medium containing 50TCID_50_ of FHV-1 and varying drug concentrations at 37 °C. Following a 48-h incubation period, cells in the positive control group displayed complete CPE when observed under the microscope. The percentage of cytopathic effect (CPE) in the cell group treated with the drug was observed and documented using an inverted microscope. The 50% effective concentration (EC50) of the drug against virus-induced CPE was calculated using the Reed–Muench method, and the curve was plotted using GraphPad Prism.

### 2.8. Plaque Reduction Assay

The CRFK cell monolayer in a 24-well plate was exposed to FHV-1 virus containing 200 plaque-forming units (PFU) and different concentrations of a drug. The exposure occurred in a cell culture incubator for 2 h. After incubation, the supernatant was removed, and 1 milliliter of low-melting-point agarose (final concentrations: 1% agarose, 2% FBS) was added to cover the cells. After 48 h, the agarose overlay was removed, and the cell monolayer was fixed with a 4% formaldehyde solution for 15 min. Following staining with 4% crystal violet, plaques were meticulously counted, and photographs were taken.

### 2.9. Quantitative Real-Time PCR

The SimplyP Virus DNA/RNA Extraction Kit (Hangzhou Bori Technology Co., Ltd., Hangzhou, China) was used to extract FHV-1 DNA. PerfectStart^®^ Ⅱ Probe qPCR SuperMix UDG (TransGen Biotech Co., Ltd., Beijing, China) and FHV-1 membrane protein (UL56) specific primers (forward primer: 5′-GCAGATCTACACATCAGA-3′, reverse primer: 5′-GGGTGATGTTAAATGTGG-3′, Probe:5′-FAM-CTCGCCTGAGATGACGGTCC-BHQ1-3′) were used for amplification. The reaction was carried out in a thermocycler (QuantStudio™ 3, ThermoFisher, Shanghai, China), according to the scheme of 5 min at 94 °C, followed by 40 cycles of 5 s at 94 °C and 30 s at 60 °C. 

For the quantitative assessment of viral nucleic acid copies, we generated positive plasmids to establish the standard curve for Fluorescent Quantitative PCR (FHV-1 qPCR). Subsequently, we employed the specified primers to conduct standard PCR amplification on the viral nucleic acid. The resulting purified PCR product was then ligated to the pMD19-T vector using T4 ligase and transfected into *E. coli* DH5α Competent Cells. Following resistance screening and sequencing, we identified strains with accurate sequencing results for large-scale cultivation. Plasmid extraction was performed, and the prepared positive plasmid was diluted to various concentrations at a tenfold ratio. Employing the specified primers and probes, we executed q-PCR, specifying “Standard” as the “Reaction Vessel Type” in Light Cycler^®^ 96 SW 1.1 software, and entered the corresponding plasmid template copy numbers. Subsequently, we selected the “Analyze Absolute Quantification” option to generate the standard curve.

### 2.10. Time of Drug Addition Assay

To investigate the mechanism of action of drugs, various drugs were introduced at different stages of viral infection in preliminary experiments [16]. These stages encompassed treatment, prevention, viral inactivation, entry, and fusion, following the procedures and incubation times outlined in Figure 1. Detection of monolayer cell infection was conducted using fluorescent quantitative PCR analysis, following the methods described previously.

### 2.11. Fusion and Entry Assays

Fusion and entry assays were conducted following the procedure outlined in Figure 1, with the variation being the utilization of DIOC and R18 for labeling FHV-1 [17] to visualize the inhibition of FHV-1 adsorption or entry into CRFK cells by Bupleurum Saikosaponin B2, Punicalin, and Punicalagin. FHV-1 was concentrated using Amicon Ultra-15 centrifugal filter units with a molecular weight cutoff of 10 kDa. The concentrated FHV-1 virus was labeled with lipophilic fluorescent dyes, DIOC and R18. DiOC and R18 were dissolved in DMSO and added to the concentrated FHV-1 virus at concentrations of 3.3 μM and 6.7 μM, respectively. The reaction mixture was gently shaken in the dark at room temperature for 1 h, followed by filtration through a 0.22 μm filter to remove unbound fluorescent dyes. This labeling allowed us to observe the inhibitory impact of the drug on either virus fusion or entry. DIOC and R18, both lipophilic dyes, played a crucial role in this study. DIOC exhibited green fluorescence upon excitation, while R18, due to its self-quenching nature at higher concentrations, emitted red fluorescence only after the virus fused with the host cell membrane, causing a reduction in its concentration. Observation of fluorescence was conducted using a Leica laser scanning confocal microscope, and quantification of fluorescence intensity was performed through ImageJ.

### 2.12. Protein Thermal Shift Assay

The Protein Thermal Shift™ dye reagent kit from ThermoFisher, USA was employed for the reaction. The procedure took place in a thermal cycler (QuantStudio™ 3, ThermoFisher) beginning at 25 °C for 2 min, followed by a gradual temperature increase at a rate of 0.05 °C/s until reaching the final temperature of 99 °C, which was maintained for 2 min.

### 2.13. Transmission Electron Microscopy (TEM)

Virus samples were incubated in a drug solution at 37 °C for 2 h. Subsequently, the prepared virus sample suspension was deposited onto copper grids coated with a thin film and allowed to stand briefly. Excess suspension was carefully removed from the edges of the copper mesh using filter paper. Negative staining was performed by adding phosphotungstic acid, a process lasting 2 min. After removing the excess staining solution with filter paper, the grid was air-dried in preparation for observation under the electron microscope.

### 2.14. Statistical Analysis

The data were expressed as mean ± standard deviation (SD) based on a minimum of three independent experiments. Statistical analysis was carried out employing GraphPad Prism 8 software. Significance levels were determined using either one-way analysis of variance (ANOVA) or Student’s *t*-test. Statistical significance was represented as follows: ns (not significant), ** p < 0.05*, *** p < 0.01*, **** p < 0.001*, and ***** p < 0.0001.*

## 3. Results

### 3.1. FHV-1 gB

The proteins were purified using the AKTA protein purification system, and their purification was confirmed through Western blot analysis (Figure 2).

### 3.2. Identification of Natural Small Molecule Inhibitors against FHV-1 gB

Small molecule inhibitors targeting FHV-1 gB have the potential to become preventive and therapeutic drugs for FHV-1 infection. In this study, BLI, a real-time detection of biomolecular interactions, was utilized to explore natural compounds as potential inhibitors of FHV-1 entry. In total, 164 known antiviral small molecule drugs were screened using BLI affinity testing (Appendix A). Initially, we confirmed the capture of gB using the (Ni-NTA) probe. The results indicated an immobilization height of 8 nm for gB on the Ni-NTA probe, meeting the height requirement for subsequent protein–small molecule interaction studies (Figure 3A). 

Following BLI screening, 16 compounds were identified for their affinity to gB (Figure 3B). Due to the strong binding exhibited by these 16 drugs, additional validation of their antiviral effects was conducted.

**Figure 3 viruses-16-00231-f003:**
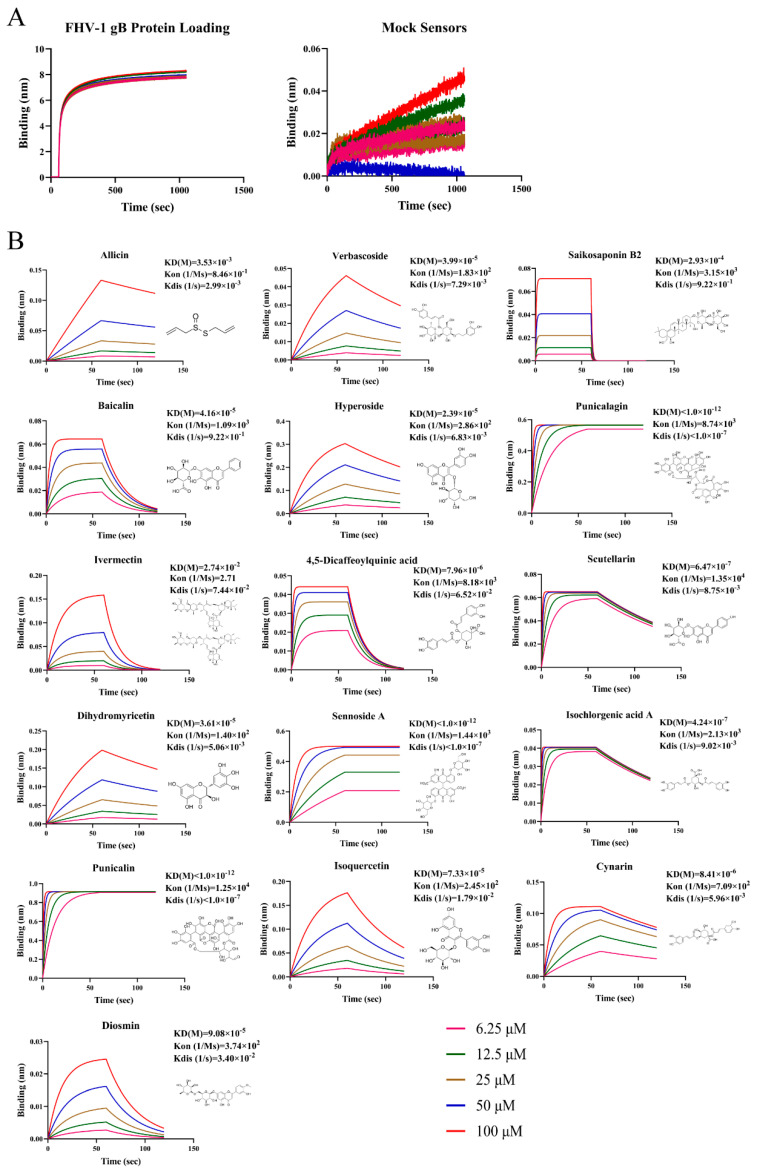
Identified natural compounds directly interact with gB through BLI binding kinetics assay. (**A**) Immobilization of gB to Ni-NTA probe and Reference Ni-NTA probe. (**B**) BLI sensorgram structures of 16 compounds showing their binding to gB. For each compound, a 1:1 fitting curve was constructed. The chemical structure and binding dissociation coefficients of the compounds are shown. The KD (M) value represents the affinity constant, Kon (1/Ms) represents the rate of association, and Kdis (1/s) represents the rate of dissociation.

### 3.3. The Natural Compounds Saikosaponin B2, Punicalin, and Punicalagin Effectively Suppress Viral Infection of FHV-1 in CRFK Cells

Among the 16 compounds exhibiting strong binding affinity, 3 compounds (Saikosaponin B2, Punicalin, and Punicalagin) were identified as effective inhibitors of FHV-1-induced cytopathic effects and plaque formation in CRFK cells (Figure 4A,B). The remaining 13 compounds showed little to no impact on inhibiting viral cytopathic effects at non-cytotoxic concentrations (Figure 5) (Table 1).

### 3.4. Saikosaponin B2, Punicalin, and Punicalagin Target the Early Viral Entry of FHV-1, Disrupting Both Viral Attachment and Entry/Fusion Events

In our study, we established a standard curve represented as  y(Cq)=−3.1702∗X (Log Quantity)+39.72 with an r^2^ value of 1. To comprehensively assess the antiviral potential of Saikosaponin B2, Punicalin, and Punicalagin, we investigated their impact on various stages of the FHV-1 lifecycle. These compounds were administered synchronously at different points during viral infection: before infection, simultaneously with infection, or after infection in CRFK cell monolayers (Figure 1). The results demonstrated that drug pretreatment did not significantly affect their antiviral efficacy against FHV-1, as evidenced by the negligible changes in FHV-1 nucleic acid copy numbers (ns) (Figure 6D). In the experimental group involving drug treatment for viral infection, a remarkably significant difference in viral nucleic acid expression was observed in the Saikosaponin-B2-treated group (*p < 0.0001*) (Figure 6E). Hence, Punicalin and Punicalagin do not exert their antiviral effects by directly acting on host cells.

To delve deeper into the underlying mechanism, we explored the drugs’ impact on specific steps of viral entry. Initially, we investigated whether the drugs directly influenced viral particle infectivity. FHV-1 particles were pre-incubated with the drugs for 80 min and then introduced onto the monolayer cells’ surface. Remarkably, the drug-treated group exhibited a clear absence of cellular CPE compared to the positive control. Moreover, a substantial difference in the viral nucleic acid copy number (*p < 0.0001*) was observed, indicating direct interference with free viral particles (Figure 6C). To assess the drugs’ effect on virus–host cell binding, incubation was conducted at 4 °C or after pre-incubation at 4 °C followed by incubation at 37 °C. Given the altered cell surface permeability at 4 °C, viral particles could only adhere to the host cell surface without undergoing fusion-related processes. In the virus fusion assay, viral nucleic acid copy numbers significantly decreased compared to the positive control group (*p < 0.0001*), signifying the drugs’ inhibition of FHV-1 binding to host cells (Figure 6B). However, in the virus entry assay, only Saikosaponin B2 exhibited inhibitory effects on FHV-1 entry into host cells at a concentration of 50 μg/mL (Figure 6A).

To visualize the early stages of viral infection and the inhibitory effects of the drugs, FHV-1 was labeled with DIOC and R18 for entry and fusion assays. Consistent with our fluorescent quantitative PCR findings, the results indicated that all three drugs exert their antiviral activity by targeting the early stages of viral infection (Figure 7A–D). In summary, our data confirm that Saikosaponin B2, Punicalin, and Punicalagin specifically target the early stages of FHV-1 infection.

**Figure 6 viruses-16-00231-f006:**
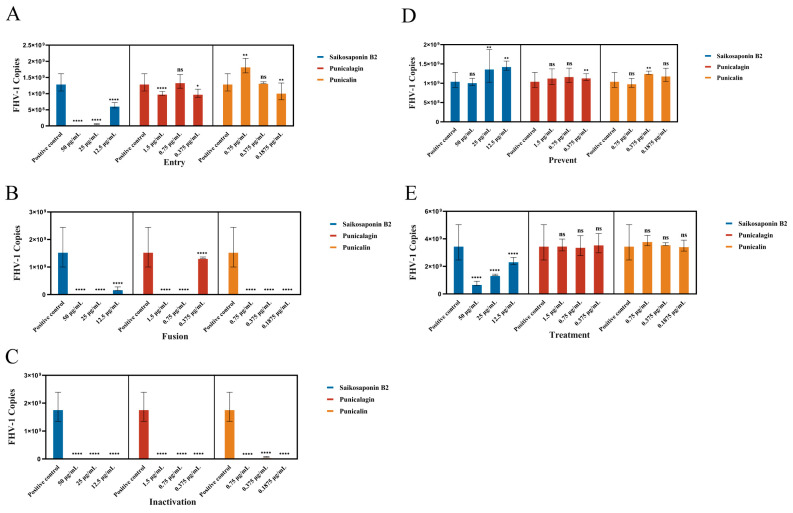
Saikosaponin B2, Punicalin, and Punicalagin target the early stages of FHV-1 entry. (**A**–**C**) Effect of Saikosaponin B2, Punicalin, and Punicalagin on free virus particles, virus entry, and viral fusion. (**D**,**E**) Saikosaponin B2, Punicalin, and Punicalagin were used to pretreat CRFK cells or applied to CRFK cells infected with FHV-1. Data are representative of three independent experiments. Data were analyzed using Student’s *t*-test (***, *p < 0.05*; ****, *p < 0.01*; *****, p < 0.0001*).

**Figure 7 viruses-16-00231-f007:**
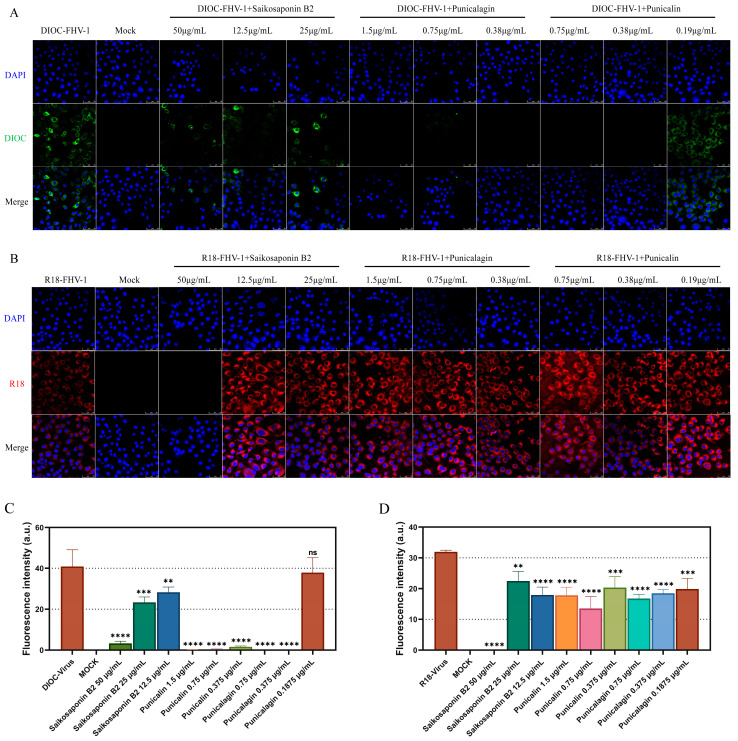
Saikosaponin B2, Punicalin, and Punicalagin target the early stages of FHV-1 entry/fusion. (**A**) Saikosaponin B2, Punicalin, and Punicalagin inhibit the adsorption of FHV-1. FHV-1 was labeled with DIOC using 10 MOI and whether the drug was added or not and incubated in CRFK cells at 4 °C for 2 h, formaldehyde-fixed and DAPI-stained, and DIOC fluorescence was detected through laser confocal microscopy. (**B**) Saikosaponin B2 inhibited the entry of FHV-1. FHV-1 was labeled with R18 using 10 MOI and whether the drug was added or not. The cells were incubated in CRFK cells at 4 °C for 2 h and then continued to be incubated at 37 °C for 2 h. The cells were formaldehyde-fixed and stained with DAPI, and the R18 fluorescence was detected through laser confocal microscopy. (**C**,**D**) Quantification of fluorescence intensity was carried out using ImageJ. Data are representative of three independent experiments. Data were analyzed using Student’s *t*-test (****, *p < 0.01*; *****, *p < 0.001*; ******, *p < 0.0001*).

### 3.5. The Drug Does Not Destroy the Structural Integrity of the Virus Particles

To further confirm the interaction between Saikosaponin B2, Punicalin, and Punicalagin and gB, we conducted a Protein Thermal Shift assay and Transmission Electron Microscopy (TEM). The results revealed a significant increase in the thermal stability of gB within the drug-treated groups compared to the negative control, indicating the drugs’ ability to interact with gB and enhance the gB protein’s stability (Figure 8). In addition, envelope proteins of FHV-1 virus particles were observed under Transmission Electron Microscopy (TEM) in the drug-treated group (Figure 9). This observation suggests that these drugs can interact with viral particles while potentially not compromising their structural integrity.

## 4. Discussion

FHV-1, initially discovered and isolated in 1958, now poses a global health threat to feline species. Current clinical treatment for FHV-1 primarily relies on a combination of nucleoside analogs and doxycycline [18]. However, these treatments involve extended courses, which can be challenging for immunocompromised kittens to adhere to. Additionally, nucleoside analogs, originally designed for HSV infections, are suboptimal for treating FHV-1 infections due to their single antiviral mechanism, making them susceptible to inducing viral resistance [19]. Consequently, the development of novel drugs offers an alternative approach to preventing and treating FHV-1 infections. Natural small molecule drugs, characterized by their compact structure, diverse target actions, and ease of synthesis, hold promise as a new class of anti-FHV-1 drugs.

Our study initially screened 16 drugs from a pool of 164 natural small molecule compounds, all with binding affinity to the FHV-1 gB protein. Among these 16 drugs, only Ivermectin is known to exhibit in vitro activity against Bovine herpesvirus 1 (BoHV-1) [20]. However, subsequent experimental results indicate that Ivermectin does not possess in vitro antiviral activity against FHV-1 within non-cytotoxic concentration ranges. Subsequent experiments confirmed the efficacy of Saikosaponin B2, Punicalin, and Punicalagin in inhibiting FHV-1. Furthermore, this research contributes significant evidence to the exploration of small molecule drugs as potential inhibitors in the context of feline herpesvirus entry.

Among drugs known for their in vitro anti-FHV-1 activity, the following EC_50_ values have been reported for CRFK cells: Ganciclovir (1.33 μg/mL), Cidofovir (3.07 μg/mL), Penciclovir (3.50 μg/mL), Foscarnet (44.71 μg/mL), Idoxuridine (1.52 μg/mL), and Acyclovir (13.04 μg/mL) [3,21]. In our tests, the EC_50_ values for the drugs were as follows: Saikosaponin B2 (13.50 μg/mL), Punicalin (0.22 μg/mL), and Punicalagin (2.19 μg/mL). These values suggest the potential of these screened drugs for clinical treatment of FHV-1 infection.

Saikosaponin B2, classified as a triterpenoid, has exhibited antiviral activity against a wide range of viruses, including the influenza A virus, hepatitis C virus, hepatitis B virus, measles virus, Severe Acute Respiratory Syndrome Coronavirus 2 (SARS-CoV-2), and respiratory syndrome virus. Additionally, it possesses the ability to inhibit the expression of various inflammatory factors, such as TNF-a, IL-6, and NF-kB [22,23,24,25]. Saikosaponin B2 is widely utilized in the treatment of lung diseases and demonstrates no adverse effects on animal health or body weight when administered through injection. Punicalin and Punicalagin, both derived from pomegranate extracts, are structurally polyphenolic compounds, and they exhibit antiviral activities similar to Saikosaponin B2. They have demonstrated effectiveness against the hepatitis C virus, SARS-CoV-2, and influenza virus [26,27,28,29]. Additionally, these compounds have shown the capacity to reduce inflammatory damage, with animal models revealing no adverse effects on the health of rats [30,31]. Given the strong antiviral effects of Saikosaponin B2, Punicalin, and Punicalagin, their ability to inhibit the expression of inflammatory factors in vitro, and their demonstrated safety in animal experiments, these three drugs emerge as promising candidates for the treatment of FHV-1 infections. Clinical trials are warranted to further evaluate their efficacy in clinical settings.

It is worth noting that although all three drugs exert their anti-FHV-1 activity by targeting gB proteins, the results from BLI revealed distinct binding and dissociation kinetics. Specifically, Saikosaponin B2 exhibited rapid binding and dissociation, while Punicalin and Punicalagin demonstrated rapid binding with no dissociation. One plausible explanation is that Saikosaponin B2 can interact with gB proteins regardless of their conformation or prior binding to cell surface receptors. In contrast, Punicalin and Punicalagin may act as competitive receptors or irreversibly lock gB proteins into closed structures, impeding their interaction with cell surface receptors. This phenomenon aligns with our experimental findings, wherein all three drugs effectively inhibited virus adsorption in both virus adsorption and inhibition assays. However, once the virus had adsorbed into the cell surface, Punicalin and Punicalagin showed reduced effectiveness in inhibiting virus infection. Therefore, we posit that the structural characteristics of these three drugs hold promise for the future development of anti-FHV-1 derivatives and offer valuable structural insights for further research on viral inhibitors.

While the screened drugs exhibit strong anti-FHV-1 activity in vitro, the lack of in vitro animal test data adds complexity to the actual scenario. Further validation is necessary to determine whether these three drugs consistently perform in different cell lines or animal models. Additionally, the potential presence of toxic side effects or safety concerns requires thorough consideration. In the upcoming phases, we can delve into the specific mechanisms by which these drugs inhibit the virus, deepening our understanding of their mode of action and laying the groundwork for designing more effective drugs in the future. Expanding our research to in vivo experiments, particularly by employing appropriate animal models, will allow for better simulation of real infection conditions. This helps verify the reliability of in vitro experimental results.

In conclusion, our study has identified Saikosaponin B2, Punicalin, and Punicalagin as potent antiviral drugs targeting the early entry of FHV-1. These drugs exert their effects by interacting with the viral surface functional protein gB. This finding holds significant implications and further underscores their potential as promising candidates for the development of antiviral drugs to treat FHV-1 infections. The discovery of these compounds highlights the importance of exploring novel therapeutic strategies and emphasizes the value of targeting the early stages of viral entry for effective antiviral interventions.

## Figures and Tables

**Figure 1 viruses-16-00231-f001:**
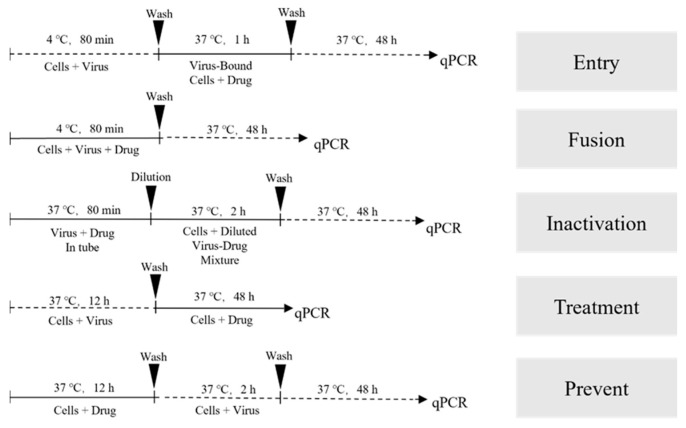
Schematic diagram of the experimental program.

**Figure 2 viruses-16-00231-f002:**
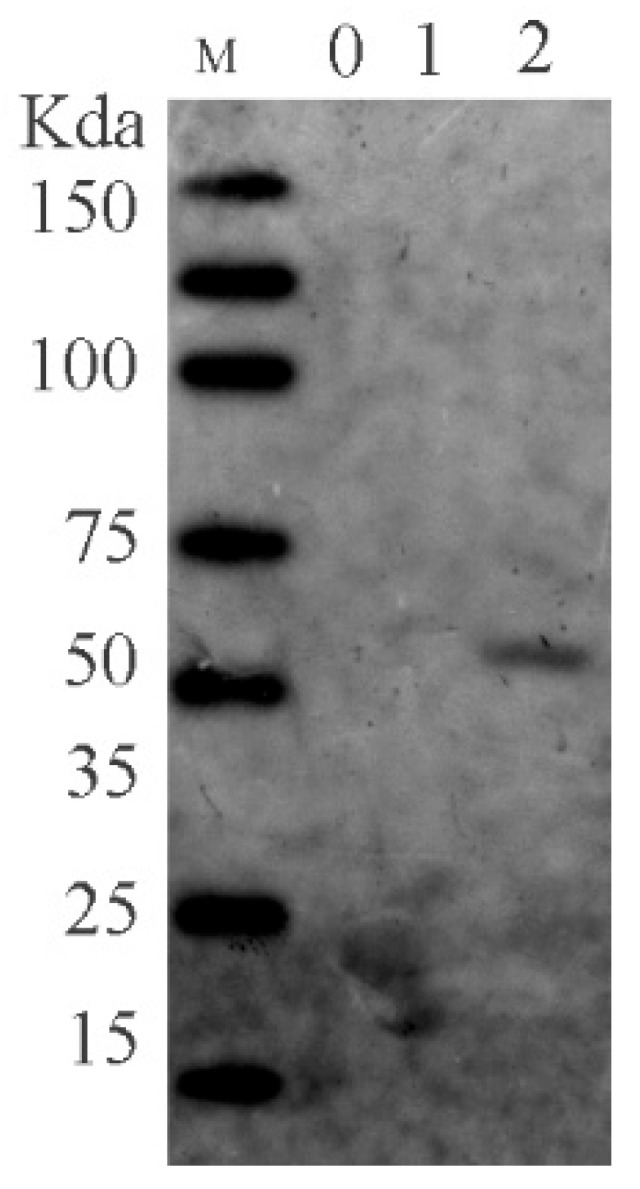
Image of FHV-1 gB Western Blot.

**Figure 4 viruses-16-00231-f004:**
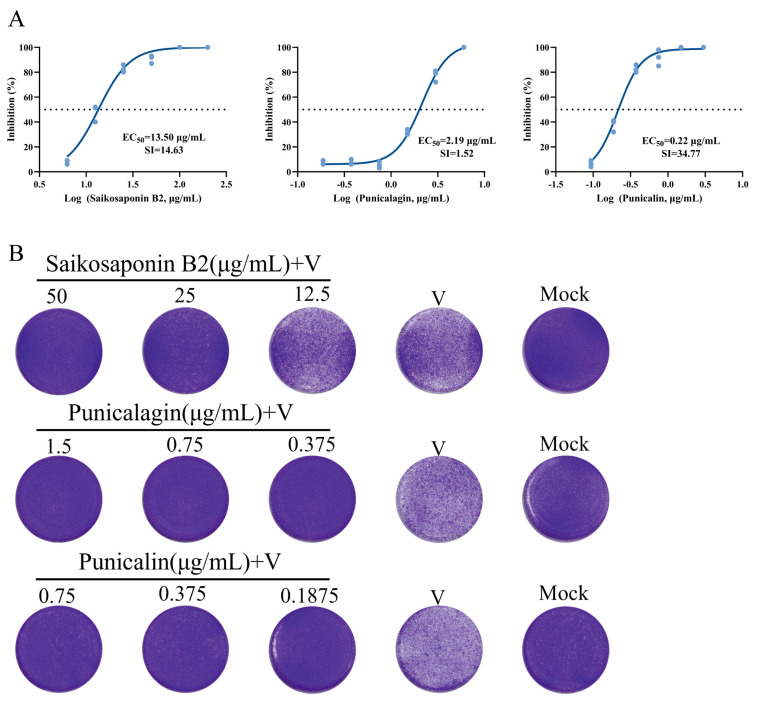
Antiviral activity of Saikosaponin B2, Punicalin, and Punicalagin against FHV-1 in CRFK cells. (**A**) Dilute the drug in cell maintenance medium containing 50 TCID_50_ FHV-1. Add the diluted drug to a 96-well plate with cells that have grown into a monolayer, with each well receiving 100 μL. In the cell culture incubator, observe cytopathic effects (CPE) in 80% of cells in the positive control group. Examine the condition of each well under an inverted optical microscope and calculate and plot the inhibition curve using GraphPad Prism. (**B**) Dilute the drug in serum-free cell culture medium containing 50 TCID_50_ FHV-1. Add the diluted drug to a 24-well plate with cells that have grown into a monolayer, with each well receiving 500 μL. After 2 h of incubation in the cell culture incubator, remove the supernatant, and overlay with 1% agarose containing 2% FBS. Continue incubation in the cell culture incubator for 48 h. After fixation with 4% paraformaldehyde and staining with crystal violet, capture images using a CTL fluorescence spot reader.

**Figure 5 viruses-16-00231-f005:**
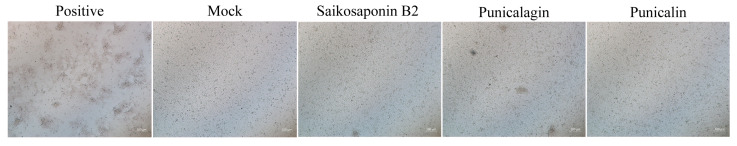
Saikosaponin B2 (50 μg/mL), Punicalin (0.75 μg/mL), and Punicalagin (1.5 μg/mL) inhibition of CPE produced by FHV-1 on CRFK cells.

**Figure 8 viruses-16-00231-f008:**
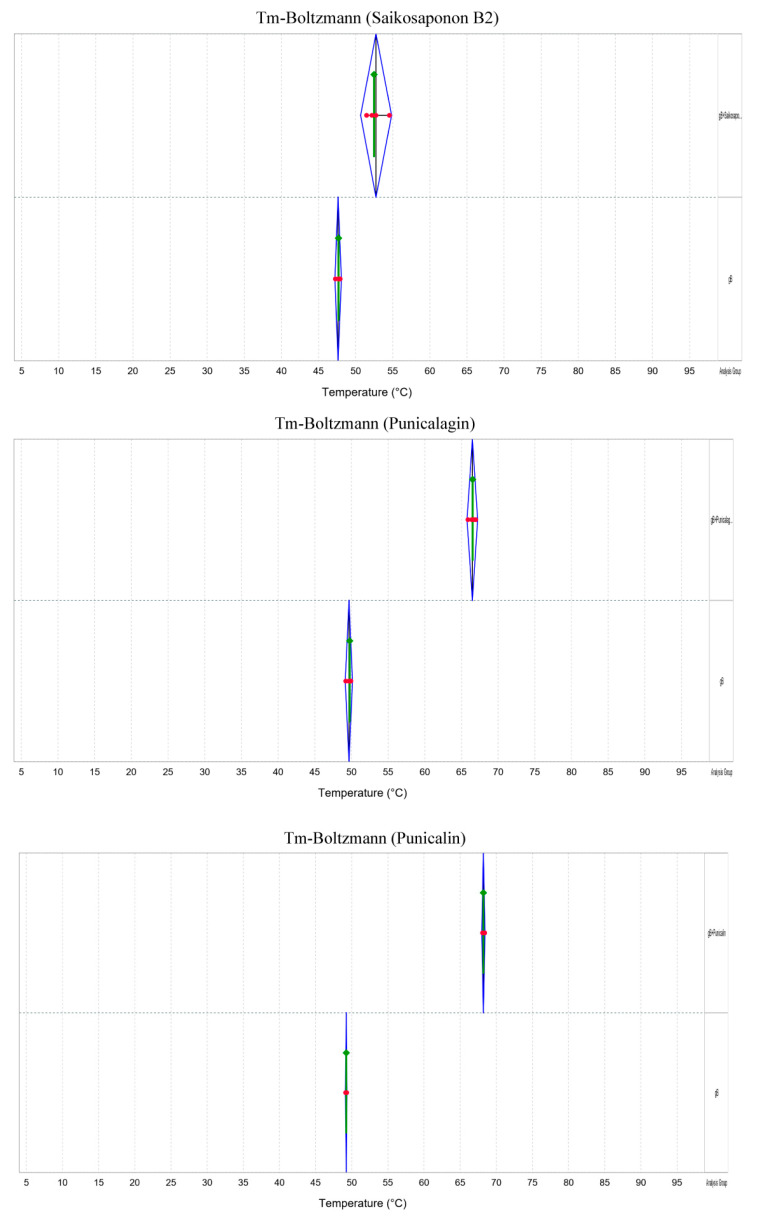
Saikosaponin B2, Punicalin, and Punicalagin altered the thermal stability of gB. Changes in Tm (Tm, the temperature at which 50% of the protein remains soluble) values of gB proteins in the presence or absence of drugs. The results showed a rightward shift in the measured Tm in the drug group compared to no drug. Each set of trials was repeated four times.

**Figure 9 viruses-16-00231-f009:**
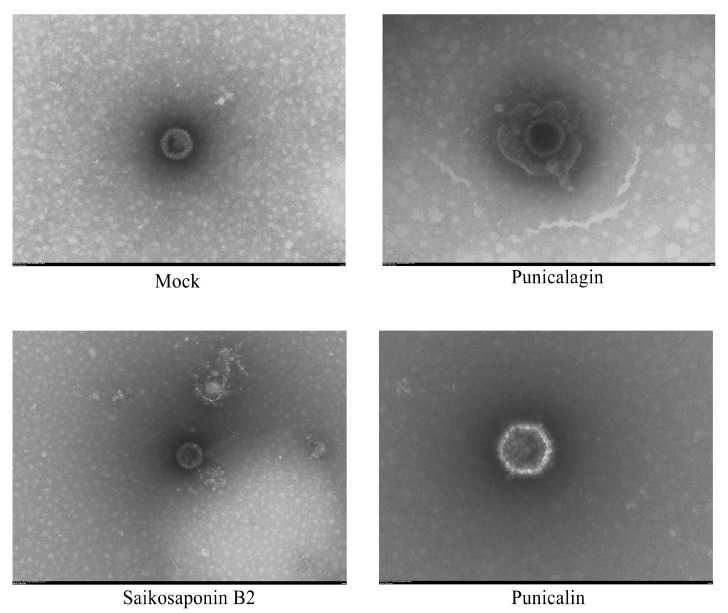
Electron microscopy of drug-treated virus particles. Saikosaponin B2, Punicalin, and Punicalagin do not disrupt the structural integrity of virus particles.

**Table 1 viruses-16-00231-t001:** Cell drug toxicity and in vitro anti-FHV-1 effect.

Combining Drugs	CC_50_ (μg/mL)	Anti-FHV-1
Saikosaponin B2	197.5 ± 0.52	√
Baicalin	65.80 ± 2.117	×
4,5-Dicaffeoylquinic acid	54.52 ± 2.89	×
Ivermectin	1.81 ± 0.06	×
Hyperoside	>50	×
Isoquercetin	289.35 ± 5.69	×
Dihydromyricetin	3.22 ± 0.43	×
Diosmin	449.75 ± 0.21	×
Verbascoside	79.56 ± 0.99	×
Allicin	84.91 ± 0.87	×
Sennoside A	130.82 ± 8.86	×
Cynarine	60.08 ± 1.86	×
Punicalin	7.65 ± 0.68	√
Punicalagin	3.32 ± 0.28	√
Scutellarin	185.2 ± 3.37	×
Isochlorgenic acid A	100.71 ± 8.45	×

CC_50_: Drug concentration at which half of the cellular activity was inhibited.

## Data Availability

The original contributions presented in the study are included in the article/supplementary material, further inquiries can be directed to the corresponding author.

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
