# Peer review of "Saikosaponin B2, Punicalin, and Punicalagin in Vitro Block Cellular Entry of Feline Herpesvirus-1"

_viruses, 2024, doi:10.3390/v16020231_

Round 1
Reviewer 1 Report
Comments and Suggestions for Authors
Feline herpes virus 1 (FHV-1) is an important pathogen of cats, and there are limited treatment options for these animals. The goal of this manuscript is to select for potential inhibitors of FHV-1 targeting glycoprotein B (gB), which is required for virus entry. Using bio-layer interferometry, the authors identified three natural compounds that bound to the FHV-1 gB protein with good affinity. They demonstrated that these compounds have very good antiviral activity and disrupt the viral attachment or entry during the early stages of infection. The authors state the virion integrity is not affected by these compounds. The authors conclude that these compounds are promising FHV-1 antivirals. The data presented in the manuscript is presented in a straightforward manner; however, certain aspects of experimental design, quantitative outcomes, and the interpretation of some data presented were unconvincing and/or lacking. Specific points are listed as below:
1. Ls. 39-41. The statement “gB is on the only glycoprotein essential for herpesvirus infection… “ is incorrect. At least one other glycoprotein is required for infection. Please correct this statement.
2. Page 2, ls.82-87: What is the strain of FHV-1 used in these experiments? Can the author give more information of the FHV-1 strain, other than it was isolated from cats. Can the authors provide more information about the process/methods involved in purifying gB?
3. Figure 2A: What method or assay did the authors use for their inhibition curve? TCID50? Please explain.
4. ls. 233-234: “Hence,…. do not exert their antiviral effects by directly acting on host cells.” This statement is not clear based on the data presented.
5. Fig.4B-F: What is meant as “Accounting Expression” of the y-axes of these graphs. Please use another term to describe what is being measured.
6. Fig. 5: What is the magnification of each microscopy image, as the images do not look uniform when examining the nuclei between images? Can the data in these experiments be quantified based on fluorescence, as it would strengthen the authors conclusions. Please provide these data/information.
7. Figure 7: The virion images presented are not convincing (e.g., virion envelope not observed in several of the images). Quantitative data are need to substantiate the authors observations in this experiment.
8. Have the authors tested a gB null mutant to confirm their conclusions about this compounds?
9. Please include the cytoxicity data for the compounds in the Supplement Data.
Comments on the Quality of English Language
The English quality is very good. Some minor typos, like "Fushion" in Figure 4C need to be corrected.
Reviewer 2 Report
Comments and Suggestions for Authors
The present study is a screening of drugs by using the Bio-layer interferometry (BLI) with the aim of identifying natural compounds exhibiting high binding affinity for the feline herpesvirus 1 (FHV-1) functional protein gB. The introduction appropriately establishes the background of FHV-1 infections and the current treatment challenges. Results are clearly presented, supporting the antiviral efficacy of Saikosaponin B2, Punicalin, and Punicalagin targeting the early entry of FHV-1. The language is generally clear, but a thorough proofread for minor grammatical and typographical errors is recommended.
Major concerns:
1. In the abstract, authors said “Specifically, these compounds, Saikosaponin B2, Punicalin, and Punicalagin, effectively hinder viral attachment or entry during the early stages of infection without compromising the integrity of the viral particle” (lane 23-25). It is not clear which phase of infection is affected by the three compounds.
2. The authors said “gB is the only glycoprotein essential for herpesvirus infection” (lane 39). It is wrong. Herpesvirus envelope is characterized by many glycoproteins, and five are involved in the early attachment of the virus to the host cell, and the subsequent fusion of membranes. Please correct these sentence.
3. In the Materials and Methods section, authors should provide more details, especially regarding the plaque purification (lane 82-83) by which FHV-1 has been isolated from infected felines.
4. The authors said “Cells were cultured in a 96-well plate for 48 hours” (lane 99). Why were the cells plated and treated 48 hours later in the cytotoxicity assay? How many cells were plated? A so long time could stimulate apoptosis and cytotoxicity results could be influenced.
5. Paragraph 2.5: Authors wrote that 50 TCID50 FHV-1 were incubated in the cytopathic effect inhibition assay, and subsequently added (and therefore diluted) to the compounds. Isn't the number of viral particles too few?
6. Paragraph 2.6: When are the substances added in the plaque reduction assays?
7. Paragraph 2.7: Primers specific for which protein of FHV-1?
8. Paragraph 2.9: Authors should describe better the method used.
9. Table S1: To enhance clarity, consider providing a concise rationale for the selection criteria of the compounds.
10. In Figure 2A, Saikosaponin B2 IC50 is 13.50 μg/mL, but there are many plaques at 12.5 μg/mL by observing Figure 2B. In which mode do authors explain these data?
11. Why did authors choose the concentrations indicated in Figure 3? They are not those reported in Figure 2.
12. The authors wrote “In our study, we established a standard curve represented as 𝑦(𝐶𝑞) = −3.1702 ∗ 225 𝑋 (𝐿𝑜𝑔 𝑄𝑢𝑎𝑛𝑡𝑖𝑡𝑦) + 39.72 with an r2 value of 1” (lane 225-226). In which mode did authors obtain it?
13. Panel 4 should be edited since there is no concordance between the figures and the text.
14. Authors said “This observation suggests that the drugs did not disrupt the viral envelope through their interaction with gB, indicating that the early antiviral effects were not mediated by the drugs' direct action on gB” (lane 280-282). It is not clear what authors would demonstrate, and Figures 6 and 7 are not very explicative.
15. There are no positive controls used in all the experiments. Please add them.
Minor concerns:
1. Be careful with abbreviations and scientific nomenclature: FHV-1, gB, SARS-CoV-2, DMSO, D-PBS need to be first mentioned as full name (feline herpesvirus type 1) followed by the abbreviation in brackets (FHV-1). Then, the short form must be used throughout the text. Viral families names need the italics.
2. Authors wrote “Among them, Ivermectin is effective against Bovine herpesvirus 1 (BoHV-1)” (lane 183-184). In which mode did authors demonstrate this data? Is it bovine or feline herpesvirus?
3. Figure 4A should be inserted in Materials and Methods and it should be enlarged.
4. Discussing study limitations and proposing future research directions would strengthen the conclusion section.
5. Cytotoxicity data are missing. Please add them is supplementary materials, at the least for 16 drugs with the higher affinity for gB.
Comments on the Quality of English Language
The language is generally clear, but a thorough proofread for minor grammatical and typographical errors is recommended.
Reviewer 3 Report
Comments and Suggestions for Authors
Lo studio dal titolo "Saikosaponin B2, Punicalin, and Punicalagin Block Cellular Entry of FHV-1 di Bin Liu et al è uno studio molto interessante perché prende in considerazione nuovi farmaci di origine naturale che potrebbero essere un'alternativa all'uso estensivo di farmaci analoghi nucleosidici che hanno innescato l'insorgenza di farmacoresistenza, diminuendo significativamente l'efficacia del trattamento nel corso degli anni. Sebbene il lavoro sia interessante, ci sarebbe bisogno di migliorarlo. Sarebbe opportuno che la parola "in vitro" comparisse nel titolo. Nell'introduzione, amplia la discussione sui farmaci a piccole molecole. Cosa sono? Saikosaponin B2, Punicalin, 25 e Punicalagin sono farmaci a piccole molecole? I farmaci utilizzati non sono nemmeno menzionati nei materiali e nei metodi. Sarebbe opportuno introdurre un paragrafo intitolato "composti" prima di "sostanze chimiche e reagenti". Alcuni acronimi non sono mai stati spiegati (ad esempio NMR, UPLC-UV, TCID50). Perché non hai usato un antimicotico oltre ai vari antibiotici? Quanto titolo ha utilizzato il virus FHV nello studio? La preparazione della glicoproteina B (gB) deve essere inclusa nei materiali e nei metodi. Il test di citotossicità non è chiaro. Le concentrazioni dei farmaci utilizzati nel test di citotossicità non sono mai spiegate. I risultati sono chiari, ben supportati dalle cifre ma non dal testo dei materiali e dei metodi.
Round 2
Reviewer 1 Report
Comments and Suggestions for Authors
The authors have done a good job in addressing most of the concerns from their previous submission. The method describing protein purification is still generally vague and should be reasonably detailed, as using purified gB is the basis for the authors' in vitro screening assay. The data regarding the electron microscopy are not convincing in their current form, unless significantly better images can be provided.
Comments on the Quality of English LanguageThe english quality is fine and performing an additional proofreading for english should be sufficient.
Reviewer 2 Report
Comments and Suggestions for Authors
Authors deeply improved the quality of the manuscript, addressing several concerns from their previous submission. However, the experimental part is still missing of essential information.
- The method describing protein purification is still very generic and should be detailed further. In addition, the TEM images are not convincing and better images should be provided.
- Authors should specify the rationale of the compounds used in the initial screening, as they reported in point-at-point response. They should add also refs corroborating this choice.
- Despite the authors’ explanation, Figure 3 is still unintelligible. Please report in the Figure caption the differences among the treatments performed (A and B indicate different results. Then do they refer to different treatments?)
